# Expression of *SCD* and *FADS2* Is Lower in the Necrotic Core and Growing Tumor Area than in the Peritumoral Area of Glioblastoma Multiforme

**DOI:** 10.3390/biom10050727

**Published:** 2020-05-07

**Authors:** Jan Korbecki, Klaudyna Kojder, Dariusz Jeżewski, Donata Simińska, Maciej Tarnowski, Patrycja Kopytko, Krzysztof Safranow, Izabela Gutowska, Marta Goschorska, Agnieszka Kolasa-Wołosiuk, Barbara Wiszniewska, Dariusz Chlubek, Irena Baranowska-Bosiacka

**Affiliations:** 1Department of Biochemistry and Medical Chemistry, Pomeranian Medical University in Szczecin, Powstańców Wlkp. 72, 70-111 Szczecin, Poland; jan.korbecki@onet.eu (J.K.); d.siminska391@gmail.com (D.S.); chrissaf@mp.pl (K.S.); rcmarta@wp.pl (M.G.); dchlubek@pum.edu.pl (D.C.); 2Department of Anaesthesiology and Intensive Care, Pomeranian Medical University in Szczecin, Unii Lubelskiej 1, 71-252 Szczecin, Poland; klaudynakojder@gmail.com; 3Department of Neurosurgery and Pediatric Neurosurgery, Pomeranian Medical University in Szczecin, Unii Lubelskiej 1, 71-252 Szczecin, Poland; djezewski@wp.pl; 4Department of Applied Neurocognitivistics, Unii Lubelskiej 1, Pomeranian Medical University in Szczecin, 71-252 Szczecin, Poland; 5Department of Physiology, Pomeranian Medical University in Szczecin, Powstańców Wlkp. 72, 70-111 Szczecin, Poland; maciejt@sci.pum.edu.pl (M.T.); patrycja.kopytko@pum.edu.pl (P.K.); 6Department of Medical Chemistry, Pomeranian Medical University in Szczecin, Powstańców Wlkp. 72, 70-111 Szczecin, Poland; izagut@poczta.onet.pl; 7Department of Histology and Embryology, Pomeranian Medical University in Szczecin, Powstańców Wlkp. 72, 70-111 Szczecin, Poland; Agnieszka.Kolasa@pum.edu.pl (A.K.-W.); Barbara.wiszniewska@pum.edu.pl (B.W.)

**Keywords:** glioblastoma multiforme, brain tumor, stearoyl–CoA desaturase, fatty acid desaturase, fatty acid

## Abstract

The expression of desaturases is higher in many types of cancer, and despite their recognized role in oncogenesis, there has been no research on the expression of desaturases in glioblastoma multiforme (GBM). Tumor tissue samples were collected during surgery from 28 patients (16 men and 12 women) diagnosed with GBM. The effect of necrotic conditions and nutritional deficiency (mimicking conditions in the studied tumor zones) was studied in an in vitro culture of human brain (glioblastoma astrocytoma) U-87 MG cells. Analysis of desaturase expression was made by qRT-PCR and the immunohistochemistry method. In the tumor, the expression of stearoyl–coenzyme A desaturase (*SCD)* and fatty acid desaturases 2 (*FADS2)* was lower than in the peritumoral area. The expression of other desaturases did not differ in between the distinguished zones. We found no differences in the expression of *SCD*, fatty acid desaturases 1 (*FADS1),* or *FADS2* between the sexes. Necrotic conditions and nutritional deficiency increased the expression of the studied desaturase in human brain (glioblastoma astrocytoma) U-87 MG cells. The obtained results suggest that (i) biosynthesis of monounsaturated fatty acids (MUFA) and polyunsaturated fatty acids (PUFA) in a GBM tumor is less intense than in the peritumoral area; (ii) expressions of SCD, SCD5, FADS1, and FADS2 correlate with each other in the necrotic core, growing tumor area, and peritumoral area; (iii) expressions of desaturases in a GBM tumor do not differ between the sexes; and (iv) nutritional deficiency increases the biosynthesis of MUFA and PUFA in GBM cells.

## 1. Introduction

Glioblastoma multiforme (GBM) is one of the most common primary brain tumors. It is also the most invasive and undifferentiated glioma and the most malignant tumor based on the histopathological criteria defined by the World Health Organization (WHO). It is characterized by high mortality despite surgical intervention, chemotherapy, and radiotherapy. The overall survival rate and 5-year postoperative survival are very low, at 8.1 months and 9.8%, respectively [1,2].

A promising direction of research into possible therapies for GBM is the metabolism of fatty acids in cancer cells [3,4]. Particularly interesting is the action of desaturases, enzymes forming double bonds in fatty acids, which can be divided into those with the highest catalytic activity for saturated fatty acids (stearoyl–CoA desaturase (SCD) and stearoyl-CoA desaturase 5 (SCD5)) or for unsaturated fatty acids (fatty acid desaturase 1 (FADS1), fatty acid desaturase 2 (FADS2), and fatty acid desaturase 3 (FADS3)).

SCD converts saturated fatty acids (SFA) into monounsaturated fatty acids (MUFA) [5], performing Δ^9^ desaturation in fatty acids with chain lengths of 12 to 19 carbons [6], with the highest specificity for stearic acid. The expression of this enzyme is highest in the adipose tissue, liver, and brain [7,8]. This enzyme plays an important role in lipid metabolism [9], and its expression is strictly regulated by hormones that are essential for the regulation of metabolism and by nutrients themselves [10]. Its expression and activity is elevated in cases of obesity [11,12,13] and insulin resistance [14]. SCD also participates in the synthesis of fatty acids, the structural elements of cell membranes, and therefore, its expression increases in rapidly dividing cells [15]. *SCD* expression is elevated in the tumors of many cancers, including esophageal cancer [16,17], hepatocellular adenoma and carcinoma [16,18], colorectal cancer [16,17], breast cancer [17,19,20], gastric cancer [17], thyroid cancer [17,21], lung adenocarcinoma [19,22], prostate cancer [19,20], and clear-cell renal cell carcinoma [23]. Importantly, the increased tumor expression of *SCD* is associated with worse prognosis.

*SCD5* is the second Δ^9^-desaturase isoform found in humans; it also occurs in most fish, amphibians, reptiles, birds, and mammals except *Muridae* [24]. Expression of this enzyme is highest in the brain, particularly in the fetal stage, and in the pancreas, adrenal gland, and ovaries [7,8]. The properties of SCD5 are similar to those of SCD. However, an experiment on murine neuroblastoma Neuro2a cells showed that SCD5 exhibited substrate specificity only for palmitic acid but not for stearic acid [25], in contrast to observations in A375 melanoma and 4T1 breast cancer cells, where SCD5 did desaturate stearic acid [26]. Unlike SCD, SCD5 may not be responsible for increasing fatty acid synthesis during cell proliferation [26]. A change in the expression of SCD5 alters the composition of phospholipids in biological membranes, increasing the synthesis of phosphatidylcholine and cholesterolesters [25]. SCD5 is also known to play an important role in oncogenesis in breast cancer [26,27], and the expression of this enzyme increases in anaplastic thyroid carcinoma [21]. On the other hand, SCD5 acts as a tumor suppressor in melanoma, with the cancer reducing its expression [28].

FADS1 and FADS2 are responsible for the biosynthetic pathways that produce long-chain polyunsaturated fatty acids (PUFA). The expression of these desaturases is highest in the liver, brain, and adrenal glands [8,29,30]. They participate in the synthesis of arachidonic acid from γ-linolenic acid and linoleic acid [31,32,33]. By producing arachidonic acid, FADS1 and FADS2 participate in the production of prostaglandins and thus play an important role in inflammation and associated neoplastic processes [34,35]. For this reason, the expression of *FADS1* and *FADS2* is increased in some cancers, e.g., Lewis lung carcinoma [35], melanoma [35], colon cancer [36], hepatocellular carcinoma [37], and breast cancer [34]. FADS1 and FADS2 also participate in the biosynthetic pathway for n-3 family long-chain PUFA, resulting in the formation of docosahexaenoic acid (DHA) and eicosapentaenoic acid (EPA) from α-linolenic acid. These fatty acids exhibit anti-inflammatory [38] and anti-cancer properties [39,40,41].

The function of *FADS3* is unclear; the gene encodes a protein with a structure similar to that of other desaturases. Its expression is highest in fatty tissue and placenta, and it is moderate in the brain [8]. FADS3 does not directly catalyze any reactions in the synthetic pathways for DHA or arachidonic acid or in the formation of MUFA from SFA [42]. Although it has displayed Δ^13^-desaturase activity for *trans*-vaccenic acid in transfected COS-7 cells and rat hepatocytes [42], those results have been contested [43]. The function of *FADS3* remains unclear, although it appears that the expression of this gene supports the synthesis of arachidonic acid and DHA in the liver and brain by regulating the expression of *FADS1* and *FADS2* [43].

Although the expression of desaturases is increased in many types of cancer and despite their recognized role in oncogenesis, there has been no research on the role of desaturases in the formation of GBM. Therefore, the aim of this study was to analyze the mRNA expression of desaturase genes—*SCD, SCD5, FADS1*, *FADS2,* and *FADS3*—in GBM in three tumor zones: necrotic core, growing tumor area, and peritumoral area. The intention was to analyze differences in the expression of individual desaturases between the three tumor zones.

## 2. Materials and Methods

### 2.1. Patient Samples

The material used in the present study was obtained from patients with brain tumors diagnosed by neuroimaging (magnetic resonance imaging (MRI) or computed tomography (CT)), who were hospitalized in the Department of Neurosurgery and Pediatric Neurosurgery of Pomeranian Medical University in Szczecin, Poland. The present project and archiving of material was initiated in 2014 by the Department of the Biochemistry and the Department of Neurosurgery and Pediatric Neurosurgery of Pomeranian Medical University in Szczecin and concerned the engagement of purinergic receptors in GBM progression. The project was accepted by the local bioethical commission (KB-0012/96/14), and the study was conducted in accordance with the Declaration of Helsinki.

Tumor tissue samples were collected during surgery from 28 patients (16 males and 12 females) diagnosed with central nervous system (CNS) tumor and GBM (Table 1). Patients presented with symptoms resulting from increased intracranial pressure such as dizziness and nausea, and those resulting from local tumor growth, which included sensory and motor disorders as well as disorders of higher nervous functions (Table 2).

Each patient was recommended for neurosurgery following the radiological diagnosis of CNS tumors. After qualifying for surgery, patients underwent a standard anesthetic procedure (general anesthesia with endotracheal intubation). During the neuronavigation procedure, craniotomy and tumor resection were performed according to the classical method (bone removal and dura incision, tumor visualization, resection, biopsy for histopathological and molecular examination, closure of the dura, bone restoration in some patients, subcutaneous tissue and skin closure in some patients). The range of resection was determined by the extent of the tumor and its topography.

Clinical–radiological morphology made it possible to distinguish three tumor zones commonly used in the literature and clinical practice: the necrotic core (NC) is usually located in the central part of the tumor, the growing tumor area (GTA) (surrounding the necrotic core), and the peritumoral area (PA): a buffer zone between the tumor and healthy tissue, with individual foci of infiltration (Figure 1) [44].

The use of neuronavigation helped map the tumor and determine the topography of the zones. The results of 1.5 and 3 Tesla MRI were entered into the computer station of a neuronavigation device and used during the operation to determine the position of surgical instruments in relation to cancer tissue with the help of a television camera. The camera monitored the surgical movement in relation to the radiological image to a precision of 2–3 mm. This allowed safe and reliable resection in places where the image of the operating microscope was uncertain and the macroscopic tumor boundaries were difficult to distinguish. Neuronavigation during biopsy and craniotomy allowed the material to be extracted from the three separate zones. Each sample was subjected to histopathological examination to confirm the criteria of grade IV brain tumor defined by the WHO: *IDH* mutation, 1p19q codeletion, and *MGMT* gene promoter methylation.

### 2.2. Immunohistochemistry

Samples of the brain tumor (glioblastoma) were fixed in 4% buffered formalin then washed with absolute ethanol (3 times over 3 h), absolute ethanol with xylene (1:1) (twice over 1 h), and xylene (3 times over 20 min). Then, the tissues were saturated with liquid paraffin for 3 h and embedded in paraffin blocks. Using a microtome (Microm HM340E), 3–5 µm serial sections were taken, placed on polysine histological slides (Thermo Scientific, UK; cat. no. J2800AMNZ), deparaffinized in xylene, and then rehydrated in decreasing concentrations of ethanol to be used for immunohistochemical staining. In order to expose the epitopes, the sections were boiled twice in a microwave oven (700 W for 4 min and 3 min) in 10 mM citrate buffer (pH 6.0). Once cooled and washed with phosphate buffer saline (PBS), the endogenous peroxidase was blocked by a 3% solution of perhydrol in methanol, and then the slides were incubated for 30 min. at room temperature (RT) with primary antibodies (anti-SCD, monoclonal mouse antibody from Invitrogen, cat. no.: MA5-27542, final dilution 1:100; and anti-FADS2 polyclonal rabbit antibody from Invitrogen, cat. no.: PA5-48353, final dilution 1:50 according to the manufacturer’s datasheet). To visualize the antigen–antibody complex, a Dako LSAB+System-HRP was used (DakoCytomation, Code K0679) based on the reaction of avidin–biotin–horseradish peroxidase with diaminobenzidine (DAB) as a chromogen, according to the included staining procedure instructions. Sections were washed in distilled water, counterstained with hematoxylin, and prepped for mounting. For a negative control, specimens were processed in the absence of primary antibodies. Positive staining was defined microscopically (Leica DM5000B, Wetzlar, Germany) by the visual identification of brown pigmentation.

### 2.3. Cell Culture and Treatment

Human brain cells (glioblastoma astrocytoma, U-87 MG cell line) from the European Collection of Authenticated Cell Cultures (ECACC) were cultured in eagle’s minimum essential medium (EMEM), (Sigma-Aldrich, Poznań, Poland) supplemented with 10% (*v*/*v*) heat-inactivated fetal bovine serum (FBS; Gibco Limited, Poznań Poland), 2 mM l-glutamine, 1 mM sodium pyruvate (Sigma-Aldrich, Poznań, Poland), 1% non-essential amino acids (Sigma-Aldrich, Poznań, Poland), 100 U/mL penicillin (Gibco Limited, Poznań, Poland), and 100 µg/mL streptomycin (Gibco Limited, Poznań, Poland), at 37 °C in a humidified atmosphere of 95% air and 5% CO_2_. The U-87 MG cells were seeded in 6-well plates at a density of 40,000 cells/cm^2^ in full medium. After 72 h incubation (70%–80% confluence), cells were washed three times with pre-warmed phosphate buffer saline (PBS) solution (37 °C). Next, the cells were cultured for 24 h under three different conditions (control, nutrient deficiency, and necrotic). The control cells were suspended in full medium, the starved cells were grown in medium with a low concentration of l-glutamine (0.2 mM) and without sodium pyruvate (volume supplemented with PBS). For the induction of necrotic conditions, cells were incubated in medium supplemented with 200 µM of H_2_O_2_. After 24 h of incubation, U-87 MG cells were trypsinized (0.25% trypsin–ethylenediaminetetraacetic acid (EDTA) solution, Sigma-Aldrich, Poznań, Poland) from the plate. After centrifuging (25 °C, 300 G, 5 min), the supernatant was discarded and the obtained cell pellet was used for RNA analysis.

In vitro studies were performed to analyze the influence of necrotic factors on the expression of desaturases. The growing GBM tumor area contained numerous and very small necrotic niches [45,46,47,48]. Around these areas, the intensive proliferation of neoplastic cells occurs. The surroundings of these areas were also poorly vascularized, resulting in a local nutritional deficiency in the GBM cells. In order to investigate the influence of necrotic factors on the expression of the desaturases in the GBM cells in this study, the U-87 MG cells were treated with 200 µM of H_2_O_2_, which is a necrotic factor commonly used in in vitro tests [49,50]. In addition, to better reflect the conditions in which GBM cells live, we studied the effect of nutritional deficiency on the expression of desaturases in U-87 MG cells. These cells grew in medium with a low concentration of l-glutamine (0.2 mM) and without sodium pyruvate. However, there was still 1.0 g/L (5.5 mM) of glucose in the medium. Under these conditions, the concentration of substances such as mineral salts, vitamins, other amino acids or growth factors were not changed. The reduction in the concentration of the selected substances significantly reduced the concentration of substrates for fatty acid production, because in the GBM cells, glucose is converted into pyruvate, then into acetyl-CoA, and then further into fatty acids. Glutamine is converted into acetyl-CoA and then into fatty acids [51,52]. In other words, we reduced the amount of substrate for the production of fatty acids and tested the expression of the tested desaturases. This model better represents the necrotic zones in the GBM tumor than the use of toxic substances that cause necrosis.

### 2.4. Quantitative Real-Time Polymerase Chain Reaction (qRT-PCR)

Quantitative analysis of mRNA expression of *FADS1*, *FADS2*, *FADS3*, *SCD,* and *SCD5* genes was performed by two-step reverse transcription PCR (RT-PCR). Total RNA was extracted from 50–100 mg tissue samples using an RNeasy Lipid Tissue Mini Kit (Qiagen) and, for the in vitro study, from 300,000 cells using an RNeasy Mini Kit. cDNA was prepared from 1 μg of total cellular RNA in 20 μL of reaction volume using a FirstStrand cDNA synthesis kit and oligo-dT primers (Fermentas). Quantitative assessment of mRNA levels was performed by real-time RT-PCR using an ABI 7500Fast instrument with Power SYBR Green PCR Master Mix reagent (Applied Biosystems). Real-time conditions were as follows: 95 °C (15 s), 40 cycles at 95 °C (15 s), and 60 °C (1 min). According to melting point analysis, only one PCR product was amplified under these conditions. Each sample was analyzed in two technical replicates, and mean Ct values were used for further analysis. The relative quantity of a target, normalized to the levels of endogenous controls *glyceraldehyde-3-phosphate dehydrogenase (GAPDH)* and *beta-2 macroglobulin (B2M)* genes, was calculated as the fold difference (2^dCt) and further processed using statistical analysis. Data were presented as tumor tissue absolute expression.

The *GAPDH* reference gene was selected because it is considered a suitable control in research on the expression of various genes in GBM [53,54,55]. Additionally, we used also a second reference gene: *B2M* [54,55].

The following primer pairs were used: (5′-TCA TGG GTG TGA ACC ATG AGA A-3′ and 5′-GGC ATG GAC TGT GGT CAT GAG-3′) for *GAPDH*, 5′-AAT GCG GCA TCT TCA AAC CT-3′ and 5′-TGA CTT TGT CAC AGC CCA AGA TA-3′) for *B2M,* (5′-CCA ACT GCT TCC GCA AAG AC 3′ and 5′-GCT GGT GGT TGT ACG GCA TA-3′) for *FADS1*, (5′-TGA CCG CAA GGT TTA CAA CAT-3′ and 5′-AGG CAT CCG TTG CAT CTT CTC-3′) for *FADS2*, (5′-GGA GTC ATC CGT CGA GTA TGG-3′ and 5′-GGG CCG ATC AGG AAG AAG T-3′) for *FADS3*, (5′-TTC CTA CCT GCA AGT TCT ACA CC-3′ and 5′-CCG AGC TTT GTA AGA GCG GT-3′) for *SCD,* and (5′-TGC GAC GCC AAG GAA GAA AT-3′ and 5′-CCT CCA GAC GAT GTT CTG CC-3′) for *SCD5*.

### 2.5. Statistical Methods

The expression values of the desaturase genes were calculated in relation to the expression of two reference genes: *B2M* and *GAPDH*. The relative expression values for the three zones (necrotic core, growing tumor area, and peritumoral area) in each patient and their ratios were calculated, e.g., growing tumor area/necrotic core as a ratio of relative expression in growing tumor area to relative expression in necrotic core. The distribution of expression values significantly differed from the normal distribution (Shapiro–Wilk test), and therefore statistical analysis was based on non-parametric tests: the Mann–Whitney U-test for comparisons between groups of patients, Wilcoxon signed-rank test for comparisons between zones of tumor, and the Spearman rank correlation coefficient for the analysis of correlations between the expression of the tested genes in the three zones. The median and quartile values were given as descriptive statistics in tables and graphs. The statistical significance threshold was *p* < 0.05. Calculations were performed using Statistica 13 software.

## 3. Results

### 3.1. The Expression of SCD and FADS2 Was Lower in GBM Growing Tumor Area than in Peritumoral Area

The expression of *SCD* in GBM was significantly lower in the growing tumor area compared to the peritumoral area and in the necrotic core compared to the peritumoral area (Figure 2). Using the *GAPDH* gene as reference, in the peritumoral area, we observed a significantly higher (almost 4 times) expression of *SCD* than in the growing tumor area (*p* = 0.015) and almost 3 times higher expression than in necrotic core (*p* = 0.019). Using the *B2M* gene as reference, the expression of *SCD* in the peritumoral area was more than 5 times higher (*p* = 0.005) than in the growing tumor area. However, the expression of *B2M* is significantly altered in the GBM tumor, and it is significantly different from that in the brain [54,55]. For this reason, all data using this reference gene should be analyzed with caution.

The expression of brain-specific *SCD5* was also lower in the growing tumor area versus the peritumoral area and in the necrotic core versus the peritumoral area, but they were not statistically significantly for either of the reference genes.

Similar to *SCD*, *FADS1* expression was lower in the growing tumor area compared to the peritumoral area and in the necrotic core compared to the peritumoral area (Figure 3). However, the observed differences were not statistically significant in relation to either of the reference genes. The expression of *FADS2* was also lower in the growing tumor area versus the peritumoral area and in the necrotic core versus the peritumoral area, but the difference was statistically significant only when using *GAPDH* as reference: *FADS2* expression in the peritumoral area was 2 times higher than in the necrotic core (*p* = 0.027).

*FADS3* expression measured against *B2M* reference gene was lower in the growing tumor area versus the peritumoral area and in the necrotic core versus the peritumoral area, but—similarly to previous genes—the observed differences were not statistically significant. The expression of *FADS3* measured against the *GAPDH* reference gene was higher in the growing tumor area versus the peritumoral area and in the necrotic core versus the peritumoral area, but again the differences were statistically insignificant.

### 3.2. Desaturase Expression Positively Correlated within and between Individual GBM Zones Glucose

Some positive correlations were found between the expression of desaturases in each of the studied tumor zones.

In the growing tumor area, *FADS1*, *FADS2*, *SCD,* and *SCD5* expression positively correlated when using the *B2M* gene as reference (Table 3). However, the expression of *B2M* is significantly altered in the GBM tumor, and it is significantly different from that in the brain [54,55]. For this reason, all data using this reference gene should be analyzed with caution.

When using *GAPDH* as a reference gene, the results were similar, although no correlation was found between the expression of *SCD* and *SCD5* (Table 4). *FADS3* expression positively correlated only with *SCD* expression, albeit only when the *GAPDH* gene was used as reference.

The expression of desaturases also correlated in the necrotic core. Using the *B2M* gene as reference, *SCD*, *SCD5*, *FADS1,* and *FADS2* expression were positively correlated in this zone. However, *FADS3* expression positively correlated only with *SCD* and *FADS1*. Using *GAPDH* as a reference gene, positive correlations were found only for *FADS2* versus *SCD5*, and *FADS1* versus *FADS3*.

Similar correlations were found in the peritumoral area. Using the *B2M* gene as reference, positive correlations were found between *SCD*, *SCD5*, *FADS1,* and *FADS2*. *FADS3* expression positively correlated only with *FADS1* expression. Similar results were obtained using the second reference gene, *GAPDH*. The expression of *FADS1*, *FADS2*, *FADS3,* and *SCD5* were positively correlated, while *FADS1* versus *SCD* and *SCD* versus *SCD5* were not. Again, *FADS3* expression positively correlated with *FADS1* expression only.

Some positive correlations were also found in the expression of individual desaturases between the studied structures. Taking *B2M* as the reference gene, correlations were observed between the growing tumor area and peritumoral area for all studied desaturase genes. There was also a positive correlation for *FADS2* and *SCD5* expression between the necrotic core and peritumoral area. However, when using the *GAPDH* gene as reference, correlations between the growing tumor area and peritumoral area were found only for *SCD* and *SCD5*.

When using the reference gene *GAPDH*, there were very few correlations between the GBM zones. A negative correlation was found between the expression of *FADS2* in the growing tumor area and the expression of *FADS3* in the peritumoral area; a positive correlation between *FADS2* expression in the necrotic core and *SCD* in the peritumoral area; and a positive correlation between *FADS2* expression in the peritumoral area and *SCD5* expression in the necrotic core. At the same time, *FADS3* expression in the peritumoral area negatively correlated with *SCD5* expression in the growing tumor area.

There were many more correlations of different desaturase expressions between the studied tumor zones with respect to the *B2M* reference gene, as shown in detail in the table below. Most were between *SCD5* and *FADS2* expression. Correlations in the expression of these two genes were found between all structures except *FADS2* expression in the necrotic core versus *SCD5* expression in the growing tumor area.

### 3.3. No Significant Differences between the Sexes in the Expression of Desaturases

The expression of *SCD* and *SCD5*, the enzymes responsible for SFA desaturation, did not differ significantly between the sexes in the growing tumor area or peritumoral area (Table 5). Although the expression of *SCD* in the necrotic core was 6 times higher in men, the relationship was statistically insignificant (*p* = 0.09 for *B2M*, *p* = 0.105 for *GAPDH*).

The expression of PUFA-processing desaturases did not differ between the sexes (Table 6). In men, *FADS1* expression was elevated in the necrotic core (*p* = 0.155 for *B2M*) and reduced in the peritumoral area (*p* = 0.133), but not statistically significantly. In men, the ratio of *FADS1* expression in the necrotic core to its expression in the peritumoral area (NC/PZ) was significantly 3 times higher than in women (*p* = 0.021, for *B2M*). There were no statistically significant sex-related differences in the expression of *FADS2*, in contrast to *FADS3;* in women in the peritumoral area, *FADS3* expression was 4 times greater than in men (*p* = 0.021, for *B2M*). However, taking into account the second reference gene, *GAPDH*, these differences were statistically insignificant (50% higher expression, *p* = 0.526).

### 3.4. The Immunolocalization of SCD and FADS2 Desaturases in Glioblastoma Multiforme Tumor

Figure 4 and Figure 5 show the immunolocalization of SCD and FADS2. The nerve cells and glia cells show the high expression of detected desaturases: SCD and FADS2 (Figure 4A and Figure 5A; red and white arrows).

The pathological features observed in the GBM (glioblastoma) are necrotic foci (NF) surrounded by hypercellular zone (HcZ; pseudopalisades, Figure 4C and Figure 5C) and also was present specialized form of angiogenesis—microvascular hyperplasia (Figure 4D and Figure 5D; green arrows). In these pathological regions, not all glia cells expressed highly SDC and FADS2 (Figure 4B–D and Figure 5C; blue arrows).

### 3.5. The Effect on Necrotic Factors on the Expression of Desaturases in U-87 MG Cells

Nutritional deficiency increased the expression of SCD (Figure 6). The expression of this gene in U-87 MG cells was statistically significantly 5 times (*p* = 0.004) and 2 times (*p* = 0.004) higher than for *GAPDH* and *B2M*, respectively. Oxidative stress increased the expression of *SCD* by 50% (compared to *B2M*, *p* = 0.004) and 3 times (compared to *GAPDH*, *p* = 0.004). Neither of these factors resulted in a statistically significant change in the expression of *SCD5* (*p* > 0.05).

Both the nutritional deficiency and necrotic conditions increased the expression of PUFA desaturases in U-87 MG cells (Figure 7). Compared to *GAPDH*, nutritional deficiency increased the expression of *FADS2* 8 times (*p* = 0.004) and necrotic conditions 5 times (*p* = 0.004). Similar results were obtained in comparison to *B2M* (*p* = 0.004), albeit the increase in expression was two times smaller. Nutritional deficiency and necrotic conditions also increased the expression of *FADS1* 2 times compared to *B2M (p* = 0.004) and 5 times compared to *GAPDH (p* = 0.004). The results show that the studied conditions caused an increase in long-chain PUFA biosynthesis.

The studied conditions also increased the expression of *FADS3*. Compared to *B2M*, it increased twice (*p* = 0.004), while compared to *GAPDH*, it increased 5 times (necrotic conditions, *p* = 0.004) and 4 times (nutritional deficiency, *p* = 0.004).

## 4. Discussion

### 4.1. Analysis of SCD Expression in GBM Tumors

In our study of GBM tumors, the expression of the *SCD* gene—one that may play a significant role in the proliferation of rapidly dividing cells—was lower in the growing tumor area and necrotic core than in the peritumoral area. This stands in opposition to other reports. For example, the induction of proliferation through transformation using simian virus 40 caused the increased expression of this enzyme [15]. *SCD* expression is also increased in neoplasms and is associated with worse prognosis for patients [16,17,18,19,20,21,22,23,56].

Although our results regarding decreased *SCD* expression in GBM are not consistent with the results of most studies on other types of neoplasms, a reduction in *SCD* expression has also been observed in prostate cancer [57]. However, most importantly, our study was the first to examine *SCD* expression in GBM, and the discrepancies with the results of other studies may result from our comparing GBM tumors with brain tissue, where *SCD* expression is higher than in other organs [8]. It is noteworthy that lesions in the brain area with the subsequent loss of eloquent and silent neurological deficit is promoting neuroneogenesis based on stem cells as well as angio and glial cells’ growth hyperactivity [58]. The latter is to be taken under consideration analyzing the *SCD* unexpected level changes in peritumoral and tumor itself areas.

However, in GBM, the expression of *SCD* has a very significant clinical impact. Research on U87 MG cells showed that *SCD* expression was increased by the activation of epidermal growth factor receptor (EGFR) [59] and platelet-derived growth factor receptor (PDGFR) [60], which are receptors that are crucial for GBM development and are often activated in GBM tumors [22,61,62]. In another study, the increased expression of *SCD* in T98G and U87 MG glioma cells correlated with higher temozolomide (TMZ)-resistance, which contributed to the reduced survival of patients after TMZ-based chemotherapy [63]. Finally, in other cancers, *SCD* expression also positively correlates with the proliferation index, which is a factor associated with the lower survival of GBM patients [64]. There are no data that would link higher *SCD* expression to negative prognoses for glioma patients. Data published in the Human Protein Atlas (available from http://www.proteinatlas.org) show 3-year survival in 26% of patients with increased *SCD* expression in a tumor and only in 5% patients with lower SCD expression [65,66]; however, these data are statistically insignificant (*p* = 0.11).

In our study, the expression of *SCD* in the peritumoral area and in the growing tumor area did not differ between the sexes, although it was distinctly but statistically insignificantly higher in the necrotic core in men. This lack of statistically significant differences confirms the observations made by other authors researching the expression of *SCD* in the brain [67,68]. However, comparisons of *SCD* expression in the necrotic core between sexes should be performed on larger groups of patients than in our study. Further research is also required to investigate the potential influence of sex hormones on *SCD* expression and GBM cell proliferation. As androgen receptor activation enhances GBM cell proliferation [69,70]—and cell proliferation is closely related to *SCD* expression [15]—the activation of androgen receptors in men could increase *SCD* expression in GBM. In women, the activation of estrogen receptor β significantly improves patient prognosis [71,72,73]. The protein expression of SCD in neurons and glia cells of tumors showed pathological changes in the brain tissue, which indicated brain tissue necrosis.

### 4.2. Analysis of SCD5 Expression in GBM Tumors

In our study, the expression of *SCD5* did not differ between the examined areas. However, the expression of the *SCD5* gene showed a statistically insignificant decrease in the necrotic core in relation to the peritumoral area. However, due to the limited number of patients, the observed dependence requires confirmation with larger groups of patients.

SCD5 is an enzyme that is specific to the brain [8], with a well-known anti-cancer effect [26,28]. Although it does not influence proliferation intensity in melanoma, it does reduce melanoma dissemination and lung metastases by reducing intracellular acidification [26]. Experiments on Neuro2a cells showed that the increased expression of *SCD5* may disturb signaling pathways relevant to GBM development, in particular by disrupting signal transmission from EGFR [25]. However, this effect of SCD5 has not been studied on any GBM model. If it turns out that SCD5 also disrupts signal transmission from EGFR in GBM, this would indicate its anti-cancer properties, since this receptor is essential for the induction of GBM cell proliferation [61,74].

In the aforementioned experiment on Neuro2a cells, SCD5 also disturbed the activation of the canonical Wnt pathway [25] and the increased expression of SCD5 caused activation of the non-canonical Wnt pathway, in which GBM may intensify the dissemination of cancer cells [25]. Activation of the canonical and non-canonical Wnt pathways is important for the functioning of cancer stem cells in GBM [75]. SCD5 may also have an oncogenic effect in other cancers, as shown on a breast cancer model, where it increased the viability of cancer cells [26,27]. *SCD5* expression is also increased in anaplastic thyroid carcinoma, which may indicate that SCD5 has important functions in that cancer [21]. *SCD5* is also likely to play an important role in neoplastic processes in GBM. Nevertheless, in glioma, SCD5 tumors may not be significant in tumor development. By The Human Protein Atlas (available from http://www.proteinatlas.org), 3-year survival of patients with high and low *SCD5* gene expression is 11% and 4%, respectively, but the differences in these results are statistically insignificant (*p* = 0.18) [65,76].

In our study, the expression of *SCD5* did not differ between male and female patients; however, the ratio of expression in the growing tumor area versus necrotic core was statistically insignificantly higher (*p* = 0.063) in women. As no studies have been conducted on sex-related differences in the expression of this enzyme in the brain or in cancers, it is very difficult to discuss these results. Due to the high expression of *SCD5* in the brain, further studies are required to examine the regulation of its expression in healthy tissues and its significance for neoplastic cells, especially in GBM.

### 4.3. Expression of FADS1 and FADS2 in GBM Tumors

In our study, the expression of *FADS2* in the necrotic core was significantly lower than in the peritumoral area. In addition, the expression of *FADS1* in the growing tumor area was distinctly, although not statistically significantly, lower than in the peritumoral area. This stands in opposition to other reports. In other types of tumors, *FADS1* and *FADS2* exhibit oncogenic properties, for example in non-small-cell lung cancer, where the lower expression of *FADS1* is associated with worse prognosis [77]. The oncogenic properties of FADS1 have also been shown in HCA-7 (colon cancer) [78] and BxPC-3 (pancreatic cancer cells) [78]. In turn, FADS2 displayed oncogenic properties in B16 melanoma [35], LLC Lewis lung carcinoma [35], MCF7 cell line (breast cancer) [34], and HT-29 human colon cancer cells [36]. In patients with esophageal squamous cell carcinoma, a lower expression of *FADS1* was associated with improved prognosis [79]. Nevertheless, it seems that FADS1 and FADS2 in glioma tumors have anti-neoplastic properties. Based on The Human Protein Atlas (available from http://www.proteinatlas.org), the greater expression of *FADS1* and *FADS2* genes in a glioma increases the 3-year survival of patients (*p* = 0.0045 for *FADS1*, *p* = 0.030 for *FADS2*) [65,80,81].

In our study, the expression of *FADS1* and *FADS2* did not differ between sexes, which is consistent with literature on their expression in the brain [67,68,82]. This stands in opposition to other organs such as the liver [82,83], where differences were related to the effects of progesterone [84]. In our study, with *B2M* as a reference, we observed only a negligible elevation in *FADS1* expression in the necrotic core in men and a statistically insignificantly elevation of expression of *FADS1* in the peritumoral area in women. This resulted in a significantly higher ratio of *FADS1* expression in the necrotic core to peritumoral area in men. Nevertheless, there are no detailed literature data concerning the influence of sex hormones on the aforementioned sex-related differences. The increased expression of *FADS1* in the peritumoral area in women could be attributed to the influence of estradiol, which increased the expression of *FADS1* in human neuroblastoma SH-SY5Y cells [85]. In the same study, the expression of *FADS1* was reduced by testosterone. However, there is no data in the literature that could help explain the mechanisms underlying the influence of sex hormones on *FADS1* expression in the necrotic core.

### 4.4. FADS3 Expression in GBM Tumors

In our study, the expression of *FADS3* did not differ between the tumor and peritumoral area. FADS3 shows Δ^13^-desaturase activity for *trans*-vaccenic acid [42], meaning it forms conjugated linoleic acids, which act as anticancer agents by activating proliferator-activated receptor γ (PPARγ) [86,87]. Another function of FADS3 is to regulate the synthesis of arachidonic acid and DHA through changes in the expression of *FADS1* and *FADS2* [43]; however, a study on *FADS3* knockout mice showed a significant age-dependent variability of n-6 and n-3 PUFA levels in the liver and brain. Nevertheless, the function of *FADS3* in GBM is not clear and requires further research. However, data from The Human Protein Atlas (available from http://www.proteinatlas.org) show that FADS3 can enhance neoplastic processes in glioma. Patients who had a higher expression of the FADS3 gene in the tumor showed a lower 3-year survival (6%) than patients with a low expression of this gene (13%, *p* = 0.039) [65,88].

In our study, the expression of *FADS3* in the peritumoral area was significantly higher in women. However, we cannot relate this observation to any literature data, as this paper is the first to study the difference in the expression of *FADS3* between sexes. Although *FADS3* is a gene located close to the relatively well-researched *FADS1* and *FADS2* [89], and at the transcription level it can possibly can be subject to regulation together with these genes, little is known about the regulation of *FADS3* expression. In the cerebral cortex, the expression of *FADS1* and *FADS2* does not differ between sexes [67,68,82,83], while in the liver, the expression of *FADS1* and *FADS2* was higher in female rats [82,83] and human women [84], which is attributed to the action of progesterone. Nevertheless, it is not possible to determine whether the sex-related difference in *FADS3* in our study was due to a difference existing in healthy brains or the simultaneous effect of sex hormones and signal molecules secreted by the GBM tumor.

### 4.5. The effect of Necrotic Factors on the Expression of Desaturases in U-87 MG Cells

The growth of GBM tissue is not accompanied by the development of blood vessels that supply the cancer cells with nutrients. This results in the formation of nutrient-deficient areas, which causes numerous small necrotic niches [45,46,47,48]. Around these areas, the intensive proliferation of neoplastic cells occurs.

According to our analysis, necrotic factors such as nutritional deficiency increased *SCD* expression but not *SCD5*. These results may explain the occurrence of intense GBM cell proliferation around the necrotic niche [45,46,47,48]. *SCD* expression is associated with the proliferation index [64]. Thus, necrotic factors increase GBM cell proliferation and thus *SCD* expression. In turn, *SCD5* expression is not associated with cell proliferation [26]. Therefore, the expression of this gene should not be higher under nutritional deficiency, as confirmed by our results.

The examined factors (nutrient deficiency) also increased the expression of PUFA desaturation: *FADS1*, *FADS2,* and *FADS3*. Currently, there is no literature data showing the influence of oxidative stress, starvation, or nutritional deficiency on the expression of the discussed desaturases. It is only known that high-carbohydrate diets increase the protein expression of FADS2 (Δ^6^ desaturase) in the rat liver but do not affect the protein expression of FADS1 (Δ^5^ desaturase) [90]. Due to the lack of literature data, it is difficult to refer to the results of other studies.

### 4.6. Effect of Mutation in IDH1/2 Genes on the Expression of Desaturases

In glioma tumor cells, point mutations often occur in one of the alleles of the isocitrate dehydrogenase (*IDH*)1 gene in codon 132 or *IDH2* gene in codon 172 [91]. The mutated IDH catalyzes the production of an oncometabolite, 2-hydroxyglutarate, from α-ketoglutarate. An inhibitor of α-ketoglutarate-dependent dioxygenases [92,93], this compound increases the methylation of histones and DNA. In addition, 2-hydroxyglutarate inhibits the activity of prolyl hydroxylases, which are enzymes that are involved in hypoxia inducible factor α subunits (HIF-α) degradation [94]. The frequency of *IDH* mutations depends on the type of GBM. Mutation in *IDH2* occurs in only 4% of GBM cases and in II–III grade gliomas [95,96]. *IDH1* mutation occurs in primary GBM in only 2% to 16% of cases, depending on the study cited [95,96,97]. In secondary GBM, the frequency of *IDH1* mutation ranges from 50% to 85% depending on the study [96,97].

Mutation in *IDH* genes causes changes in the synthesis of fatty acids. Experiments on acute myeloid leukemia HL60 line have demonstrated that *IDH1* mutation causes an increase in lipid anabolic fluxes [98] and significantly more MUFA and more SFA, albeit not statistically significantly (*p* = 0.07) [98]. In the same experiment, the level of PUFA in the studied cells did not differ between those with or without the *IDH* mutation (*p* = 0.6) [98]. The observed changes were confirmed on the glioma tumor with and without the *IDH1* mutation. The *IDH1* mutation causes an increase in SCD activity in the tumor [99]. The mutation increases EPA and docosapentaenoic acid (DPA), but it does not affect DHA nor linoleic acid, linolenic acid, and arachidonic acid [99]. This indicates an increase in FADS1 activity, although with no effect on arachidonic acid.

Biosynthesis 2-hydroxyglutarate may cause endoplasmic reticulum autophagy, which results in the degradation of enzymes involved in lipid synthesis [100]. Thus, it may reduce the protein expression of genes tested in this study.

The study cited above shows that the mutation of *IDH1* in GBM results in an increase in *SCD* expression (increase in the amount of MUFA) and probably an increase in *FADS1* expression, but it does not affect the expression of *FADS2* (no effect on the overall PUFA level).

### 4.7. Conclusions

The obtained results suggest that (i) biosynthesis of MUFA and PUFA in a GBM tumor is less intense than in the peritumoral area; (ii) expressions of SCD, SCD5, FADS1, and FADS2 correlate with each other in the necrotic core, growing tumor area, and peritumoral area; (iii) expressions of desaturases in a GBM tumor do not differ between the sexes; and (iv) nutritional deficiency increases the biosynthesis of MUFA and PUFA in GBM cells.

## Figures and Tables

**Figure 1 biomolecules-10-00727-f001:**
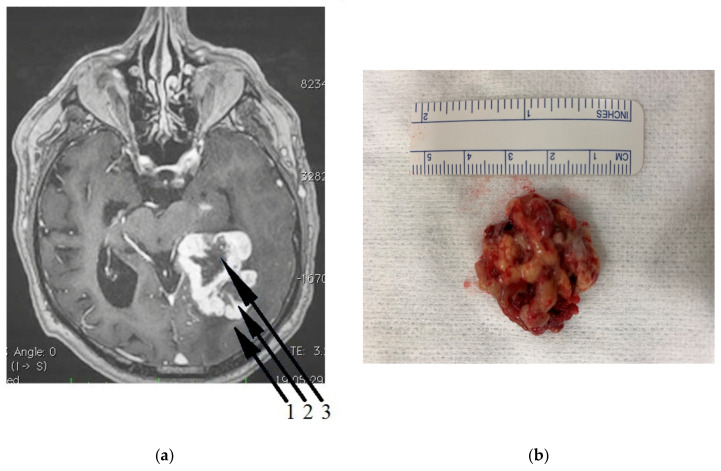
(**a**) Magnetic resonance imaging (MRI) scan of a patient with diagnosed glioblastoma multiforme (GBM). The particular zones 1,2,3 are visible on the tumor cross-section. Zone 1—the peritumoral area, Zone 2—the growing tumor area, Zone 3—the necrotic core. (**b**) GBM tissue obtained from temporal lobe. The tumor was removed by neuronavigation-guided craniotomy. II gyrus cortex was incised over a 5 cm section from the polar end and the central part of the tumor excised, followed by careful and fine debridement with an ultrasonic surgical aspirator.

**Figure 2 biomolecules-10-00727-f002:**
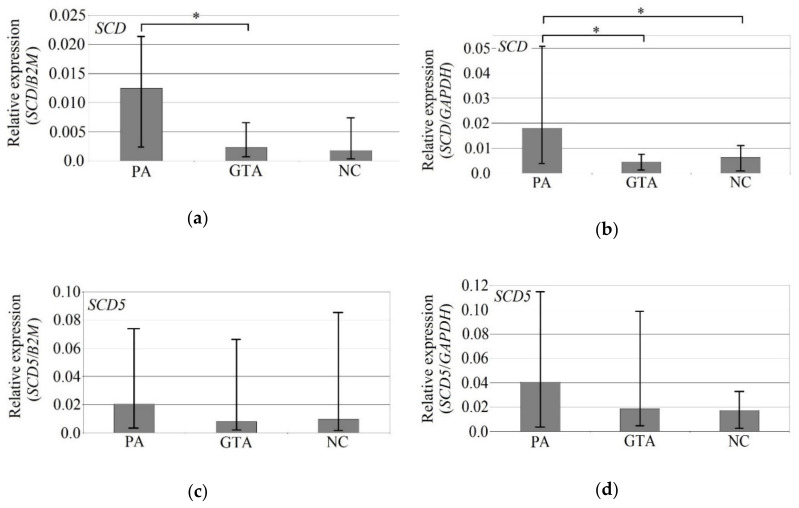
mRNA expression of stearoyl–CoA desaturase (*SCD*) (**a**,**b**) and *SCD5* (**c**,**d**) depending on the location in GBM, using *B2M* and *GAPDH* genes as reference. The graphs show the median mRNA expression of a given desaturase. The error bars are the first and third quartiles. * - statistically significant difference in the expression of a given desaturase between the tumor area according to the Wilcoxon signed-rank test (*p* < 0.05). PA—peritumoral area (n = 22); GTA—growing tumor area (n = 19); NC—necrotic core (n = 21).

**Figure 3 biomolecules-10-00727-f003:**
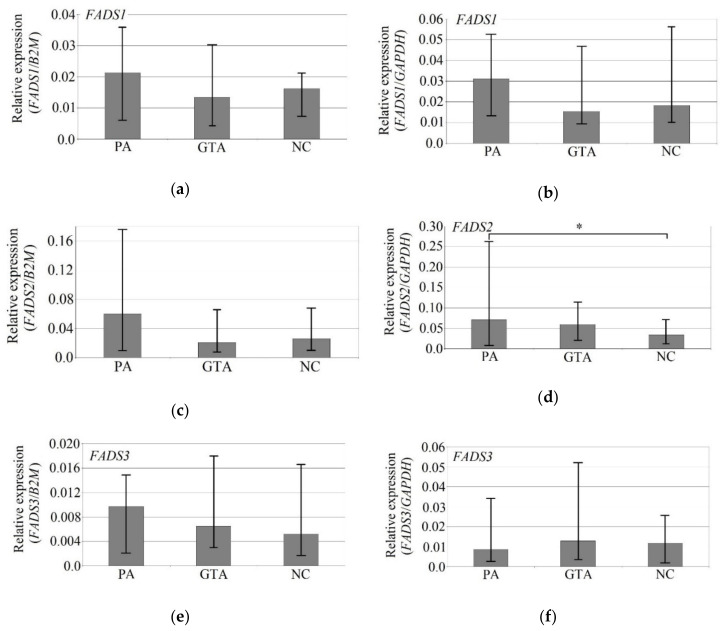
mRNA expression of *FADS1* (**a**,**b**), *FADS2* (**c**,**d**), and *FADS3* (**e**,**f**) depending on the location in GBM, using *B2M* and *GAPDH* genes as reference. The graphs show the median mRNA expression of a given desaturase. The error bars are the first and third quartiles. * - statistically significant difference in the expression of a given desaturase between the tumor area according to the Wilcoxon signed-rank test (*p* < 0.05). PA—peritumoral area (n = 22); GTA—growing tumor area (n = 19); NC—necrotic core (n = 21).

**Figure 4 biomolecules-10-00727-f004:**
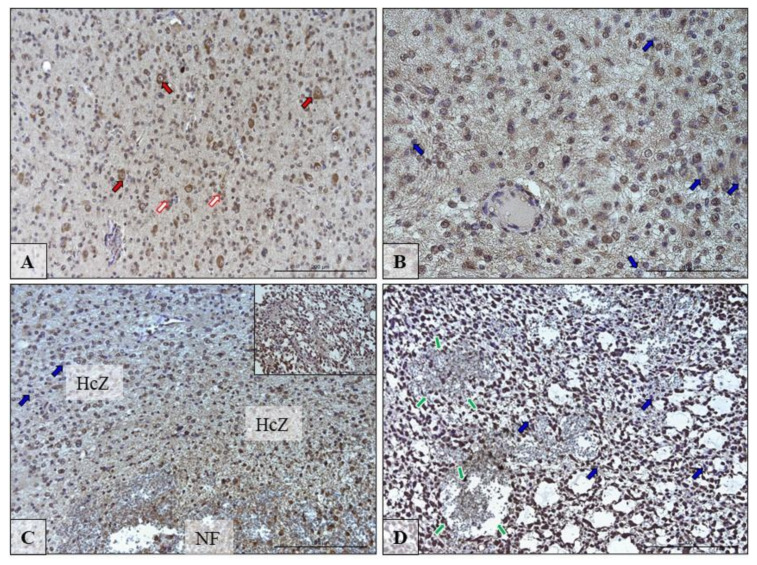
Representative microphotographs showing the protein expression of desaturase SCD in neurons and glia cells in the peritumoral area (**A**; red and white arrows) and in cells in the growing tumor area (**B**; blue arrows) necrotic core (**C**,**D**; blue arrows) of brain tumors from patients diagnosed with glioblastoma. Please note that on microphotography of GTA/NC, there are many pathologically changed regions of the brain tissue that indicate brain cell necrosis envelope by hypercellular zones (HcZ), with additional microvascular hyperplasia (green arrows). There are microphotographs from different objective (obj. magnification: **A**,**C**,**D** ×20, **B** ×40); scale bar are 200 µm or 100 µm.

**Figure 5 biomolecules-10-00727-f005:**
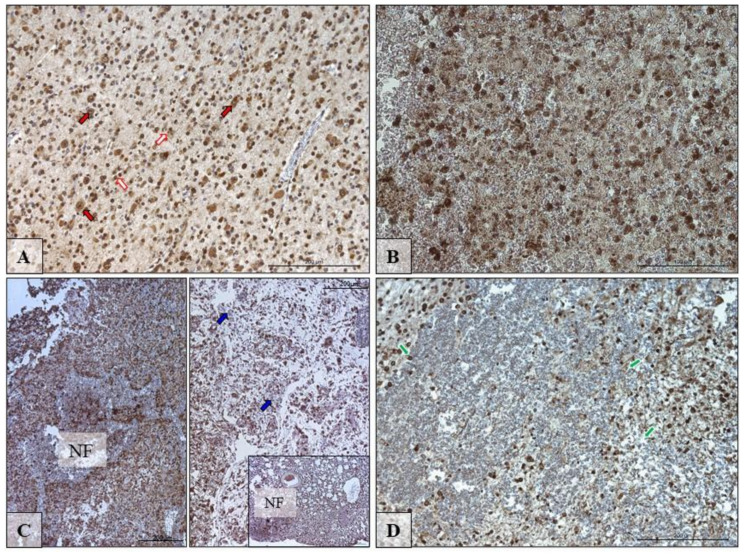
Representative microphotographs showing the protein expression of desaturase FADS2 in neurons and glia cells in the peritumoral area (**A**; red and white arrows) and in cells in the growing tumor area (**B**) necrotic foci (NF) (**C**,**D**; blue arrows) of brain tumors from patients diagnosed with glioblastoma. Please note that on microphotography of GTA/NC, there are many pathologically changed regions of the brain tissue that indicate brain cell necrosis envelope by hypercellular zones (HcZ), with additional microvascular hyperplasia (green arrows). There are microphotographs from different objective (obj. magnification: **A**, ×20, **C** ×20, **B**,**D** ×40); scale bars are 200 µm or 100 µm.

**Figure 6 biomolecules-10-00727-f006:**
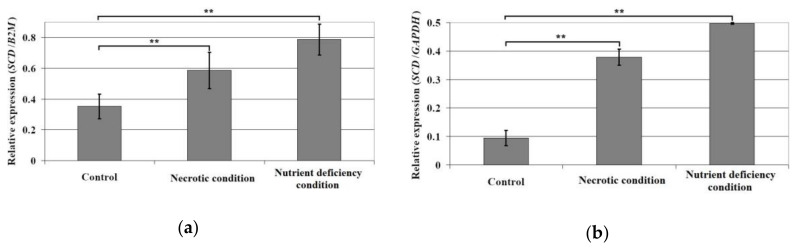
The effect of necrotic condition or nutritional deficiency on the mRNA expression of *SCD* (**a**,**b**) and *SCD5* (**c**,**d**). Human brain (glioblastoma astrocytoma) U-87 MG cell line cultured in eagle’s minimum essential medium (EMEM) with 10% fetal bovine serum (FBS), 2 mM l-glutamine, 1 mM sodium pyruvate, 1% non essential amino acids, 100 U/mL penicillin, and 100 µg/mL streptomycin. U-87 MG cells were seeded in 6-well plates. After 72 h incubation, cells were washed three times with phosphate-buffered saline (PBS) solution and cultured for 24 h under three different conditions (control, nutrient deficiency, and necrotic). Control cells were suspended in full medium, starved cells were grown in medium with a low concentration of l-glutamine (0.2 mM) and without sodium pyruvate. For the induction of necrotic conditions, cells were incubated in medium supplemented with 200 µM H_2_O_2_. After 24 h of incubation, the cells were trypsinized and analyzed using qRT-PCR. **—statistically significant difference in the expression of a given desaturase between the tumor area according to the Wilcoxon signed-rank test (*p* < 0.01).

**Figure 7 biomolecules-10-00727-f007:**
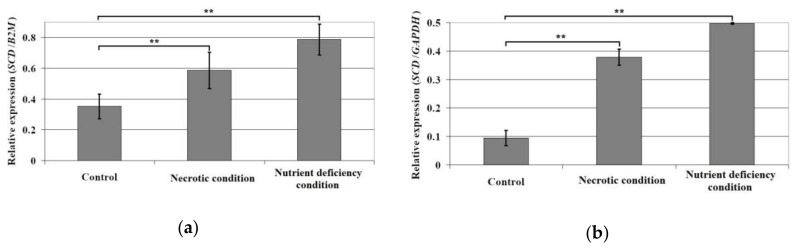
Effect of the necrotic condition or nutritional deficiency on the mRNA expression of FADS1 (**a**,**b**), *FADS2* (**c**,**d**), and *FADS3* (**e**,**f**). Human brain (glioblastoma astrocytoma) U-87 MG cell line cultured in EMEM medium with 10% FBS, 2 mM l-glutamine, 1 mM sodium pyruvate, 1% non-essential amino acids, 100 U/mL penicillin, and 100 µg/mL streptomycin. U-87 MG cells were seeded in 6-well plates. After 72 h incubation, cells were washed three times with PBS solution. Next, cells were cultured for 24 h under three different conditions (control, nutrient deficiency, and necrotic). Control cells were suspended in full medium, while starved cells were grown in medium with low concentration of l-glutamine (0.2 mM) and without sodium pyruvate. For the induction of necrotic conditions, cells were incubated in medium supplemented with 200 µM H_2_O_2_. After 24 h of incubation, the cells were trypsinized and analyzed using qRT-PCR. **—statistically significant difference in the expression of a given desaturase between the tumor area according to the Wilcoxon signed-rank test (*p* < 0.01).

**Table 1 biomolecules-10-00727-t001:** The statistical characteristics of the study group.

	N	Mean	Standard Deviation	Median	Minimum	Maximum	First Quartile	Third Quartile	Interquartile Range
Age at surgery	24	60.7	12.5	64	36	81	54	68.5	14.5
Weight	24	84	19	89	55	130	67.5	95	27.5
Height	23	172	12	172	147	196	163	182	19
Body mass index (BMI)	23	28.7	4.8	27.9	21.5	38.9	24.7	31.9	7.2
Physical activity	21	3.05	1.16	3	1	4	3	4	1
Limitation of physical activity caused by disease	21	2.10	0.83	2	1	3	1	3	2
Limitation of cognitive abilities	21	2.19	0.87	2	1	3	1	3	2

N: number of patients included in the analysis. Physical activity, limitation of physical activity, and limitation of cognitive abilities were calculated on the basis of the levels indicated in the questionnaires. Physical activity: everyday—4/a few times a week—3/rarely—2/almost never—1. Limitation of physical activity: none—1/partial—2/considerable—3. Limitation of cognitive abilities: none—1/partial—2/considerable—3.

**Table 2 biomolecules-10-00727-t002:** Inclusion and exclusion criteria. GBM: glioblastoma multiforme.

Inclusion Criteria	Exclusion Criteria
Age 18+	Age below 18
Central nervous system tumor confirmed by radiological examination	Histopathological diagnosis other than GBM
Surgical and anesthetic qualification for surgery	
Histopathological diagnosis of biopsied tumor mass: GBM	

**Table 3 biomolecules-10-00727-t003:** Correlations between the mRNA expression of all studied desaturase genes in all three tumor zones, using *B2M* gene as reference.

Studied Gene/ Location		*FADS1*			*FADS2*			*FADS3*			*SCD*			*SCD5*		
		GTA	NC	PA	GTA	NC	PA	GTA	NC	PA	GTA	NC	PA	GTA	NC	PA
*FADS1*	GTA	1.00														
	NC	0.26	1.00													
	PA	**0.48** *	0.15	1.00												
*FADS2*	GTA	**0.77** *	0.09	0.51	1.00											
	NC	0.15	**0.58** *	0.33	0.48	1.00										
	PA	0.22	0.10	**0.48** *	**0.50** *	**0.53** *	1.00									
*FADS3*	GTA	0.10	−0.11	0.21	0.18	0.08	0.40	1.00								
	NC	0.00	**0.52** *	−0.01	0.14	0.21	0.37	0.40	1.00							
	PA	−0.10	−0.03	**0.43** *	0.12	0.23	0.37	**0.71** *	0.29	1.00						
*SCD*	GTA	**0.58** *	0.16	0.13	**0.68***	0.50	0.40	0.22	0.17	0.25	1.00					
	NC	0.35	**0.63** *	0.04	0.38	**0.45** *	0.07	0.12	**0.59** *	0.09	0.30	1.00				
	PA	0.20	0.35	**0.46** *	0.46	**0.68** *	**0.74** *	0.35	**0.49** *	0.39	**0.64** *	0.40	1.00			
*SCD5*	GTA	**0.65** *	−0.07	**0.48** *	**0.82** *	0.35	**0.52** *	0.03	−0.05	−0.05	**0.56** *	0.20	**0.57** *	1.00		
	NC	0.30	**0.66** *	0.37	**0.51** *	**0.80** *	**0.65** *	0.05	0.42	0.25	**0.52** *	**0.63** *	**0.76** *	0.48	1.00	
	PA	**0.47** *	0.21	**0.55***	**0.68** *	**0.62** *	**0.70** *	0.16	0.08	−0.02	0.24	0.22	**0.59** *	**0.65** *	**0.71** *	1.00

The values of Spearman’s rank correlation coefficients are given. * statistically significant correlation of the expression of two genes/locations (*p* < 0.05). GTA—growing tumor area; NC—necrotic core; PA—peritumoral area.

**Table 4 biomolecules-10-00727-t004:** Correlations between the mRNA expression of all studied desaturase genes in all three tumor zones, using *GAPDH* gene as reference.

Studied Gene/ Location		*FADS1*			*FADS2*			*FADS3*			*SCD*			*SCD5*		
		GTA	NC	PA	GTA	NC	PA	GTA	NC	PA	GTA	NC	PA	GTA	NC	PA
*FADS1*	GTA	1.00														
	NC	0.44	1.00													
	PA	0.22	0.31	1.00												
*FADS2*	GTA	**0.59** *	−0.31	−0.20	1.00											
	NC	0.01	0.13	0.11	−0.07	1.00										
	PA	0.16	−0.26	**0.52** *	0.13	0.45	1.00									
*FADS3*	GTA	**0.47** *	0.14	−0.01	0.29	−0.16	−0.01	1.00								
	NC	0.13	**0.57** *	−0.13	−0.20	−0.18	−0.09	0.36	1.00							
	PA	−0.16	0.19	**0.53** *	−0.53 *	0.02	0.31	0.26	0.32	1.00						
*SCD*	GTA	**0.78** *	0.30	0.10	**0.67** *	0.18	0.14	**0.55** *	0.12	−0.13	1.00					
	NC	0.14	0.44	−0.06	0.10	0.24	−0.14	0.02	0.26	0.11	0.13	1.00				
	PA	0.35	−0.09	0.44	0.37	**0.50** *	**0.66** *	0.22	−0.17	0.32	**0.51** *	0.00	1.00			
*SCD5*	GTA	**0.65** *	−0.21	0.09	**0.78** *	−0.05	0.20	0.11	−0.44	−0.55 *	0.46	0.02	0.38	1.00		
	NC	0.19	0.03	0.03	0.13	**0.69** *	**0.48** *	−0.09	−0.19	0.00	0.15	0.40	0.45	0.14	1.00	
	PA	0.34	−0.19	**0.56** *	0.30	0.33	**0.72***	−0.01	−0.35	0.03	0.04	0.02	0.42	**0.62** *	0.40	1.00

The values of Spearman’s rank correlation coefficients are given. * statistically significant correlation of the expression of two genes/locations (*p* < 0.05). GTA—growing tumor area; NC—necrotic core; PA—peritumoral area.

**Table 5 biomolecules-10-00727-t005:** mRNA expression of *SCD* and *SCD5* by sex.

Sex	Female				Male				
Studied Gene/Reference Gene/Location	Median	1st Quartile	3rd Quartile	N	Median	1st Quartile	3rd Quartile	N	p
*SCD/* *B2M*									
GTA	0.0023	0.0013	0.0064	9	0.0009	0.0004	0.0033	9	0.233
NC	0.0004	0.0002	0.0017	8	0.0026	0.0004	0.0193	12	0.090
PA	0.0063	0.0018	0.0207	8	0.0104	0.0019	0.0202	12	1.000
GTA/NC	3.5421	1.0997	9.4235	8	0.2712	0.0410	1.3336	7	0.083
*SCD/GAPDH*									
GTA	0.0044	0.0008	0.0107	9	0.0043	0.0017	0.0060	9	0.965
NC	0.0012	0.0005	0.0036	8	0.0078	0.0010	0.0125	12	0.105
PA	0.0092	0.0021	0.0470	8	0.0186	0.0046	0.0468	12	0.758
*SCD5/* *B2M*									
GTA	0.0079	0.0017	0.0432	10	0.0078	0.0016	0.0088	9	0.568
NC	0.0016	0.0006	0.0102	9	0.0121	0.0036	0.0854	12	0.201
PA	0.0054	0.0021	0.0423	9	0.0235	0.0038	0.0654	13	0.664
GTA/NC	3.5979	1.3722	5.6384	9	0.3880	0.1523	0.9700	7	0.063
*SCD5/GAPDH*									
GTA	0.0090	0.0026	0.0199	10	0.0299	0.0046	0.0967	9	0.327
NC	0.0065	0.0012	0.0177	9	0.0235	0.0081	0.0328	12	0.320
PA	0.0085	0.0024	0.0502	9	0.0629	0.0090	0.1460	13	0.117

GTA—growing tumor area; NC—necrotic core; PA—peritumoral area. GTA/NC—the ratio of expression in growing tumor area to expression in necrotic core.

**Table 6 biomolecules-10-00727-t006:** mRNA expression of *FADS1*, *FADS2*, and *FADS3,* by sex.

Sex	Female				Male				
Studied Gene/Reference Gene/Location	Median	1st Quartile	3rd Quartile	N	Median	1st Quartile	3rd Quartile	N	p
*FADS1/* *B2M*									
GTA	0.0175	0.0051	0.0271	10	0.0065	0.0039	0.0143	9	0.369
NC	0.0128	0.0069	0.0175	9	0.0208	0.0054	0.0402	12	0.155
PA	0.0277	0.0155	0.0357	9	0.0088	0.0021	0.0306	13	0.133
NC/PA	0.4949	0.2562	0.6068	8	1.7582	0.4924	4.4001	11	0.021 *
*FADS1/GAPDH*									
GTA	0.0121	0.0050	0.0204	10	0.0394	0.0112	0.0591	9	0.142
NC	0.0115	0.0035	0.0485	9	0.0183	0.0116	0.0548	12	0.477
PA	0.0340	0.0084	0.0497	9	0.0286	0.0050	0.0513	13	0.867
*FADS2/* *B2M*									
GTA	0.0205	0.0086	0.1085	10	0.0153	0.0044	0.0392	9	0.414
NC	0.0179	0.0075	0.0468	9	0.0260	0.0024	0.1177	12	0.670
PA	0.0836	0.0130	0.2199	9	0.0448	0.0079	0.1201	13	0.193
*FADS2/GAPDH*									
GTA	0.0482	0.0116	0.0644	10	0.0485	0.0184	0.1061	9	0.744
NC	0.0303	0.0066	0.0493	9	0.0301	0.0112	0.1144	12	0.887
PA	0.0163	0.0096	0.2084	9	0.0867	0.0057	0.2571	13	0.973
*FADS3/* *B2M*									
GTA	0.0086	0.0053	0.0178	10	0.0029	0.0019	0.0057	9	0.060
NC	0.0020	0.0013	0.0048	9	0.0073	0.0017	0.0242	12	0.201
PA	0.0112	0.0096	0.0255	9	0.0030	0.0010	0.0094	13	0.021 *
NC/PA	0.2007	0.1072	0.2872	8	3.1901	0.5512	9.2635	11	0.013 *
*FADS3/GAPDH*									
GTA	0.0100	0.0026	0.0280	10	0.0205	0.0032	0.0474	9	0.568
NC	0.0068	0.0008	0.0226	9	0.0111	0.0029	0.0257	12	0.356
PA	0.0103	0.0022	0.0358	9	0.0067	0.0032	0.0255	13	0.526

*—statistically significant difference between two groups according to the Mann–Whitney U test (*p* < 0.05). GTA—growing tumor area; NC—necrotic core; PA—peritumoral area; NC/PA—the ratio of expression in necrotic core to expression in peritumoral area.

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
