# Peer review of "Expression of SCD and FADS2 Is Lower in the Necrotic Core and Growing Tumor Area than in the Peritumoral Area of Glioblastoma Multiforme"

_biomolecules, 2020, doi:10.3390/biom10050727_

Round 1

Reviewer 1 Report

Authors improved the manuscript, however they only perform immunohistochemistry for SCD, but not for FASD1, FASD2 and FASD3. They also introduced two other paragraphs that were interesting in the context of the paper. 

Author Response

Thank you very much for review. According to Reviewer remark we performed IHC study of SCD expression in all three tumor areas. In addition, we performed an expression analysis of FADS2. In PCR studies, gene expression only for these proteins was reduced in GBM tumor. For this reason, results obtained with immunocytochemistry method complement the results obtained. Both proteins are also the most important of all fatty acid desaturases. SCD is the main enzyme involved in MUFA synthesis. In contrast, FADS2 is an enzyme in the first stage of the long chain poly-unsaturated fatty acids biosynthesis pathway. For this reason, it is the most important enzyme regulating the synthesis of this pathway.
We can perform IHC analysis of other desaturases, but we do not know when we can do it. Our laboratory is in a hospital transformed into a unit in which patients with confirmed covid-19 are isolated. We had purchased anti-SCD and anti-FADS2 antibodies for other projects that we used to perform the analysis for this work. However, as the epidemic progresses, we already have a closed supply department, and a large staff work remotely. Mainly prepares already obtained results from already performed experiments. From statistics I can see that the situation in our country (Poland) is stable, i.e. for a long time the daily number of new infected people has been similar. This state will last optimistically counting a month. Another month until all patients recover and cancel the epidemic. Of course, with pessimistic predictions, we may have these problems until autumn. Then, in turn, there will be an increase in the incidence of covid-19 if this new virus is similar to other coronaviruses, causing 15% of all seasonal colds.
For this reason, we are currently unable to order antibodies against other desaturases in the current situation. We can do it but we don't know if in 2 months or at a later date. Nevertheless, we believe that the immunocytochemistry performed sufficiently complements our qRT-PCR results. We performed an analysis of the most important desaturases. At the same time, we analyzed all desaturases whose expression is significantly different between the tumor and the peritumoral area.

Reviewer 2 Report

Thanks for your effort on the final revision of your manuscript except one thing.

The authors have to add conclusion  as the conclusion of Abstract in the conclusion part of the Discussion section. For example, the results suggest that ........

Author Response

Thank you very much for review. According to Reviewer remark we corrected conclusion and Abstract.

This manuscript is a resubmission of an earlier submission. The following is a list of the peer review reports and author responses from that submission.

Round 1

Reviewer 1 Report

In this review manuscript, authors described on ‘Expression of SCD and FADS2 is lower in the necrotic core and growing tumor area than in the peritumoral area of glioblastoma multiforme”. Although the results are partially original and interested in readers, there are many weakness and flaws that require authors’ further attention:         

Major points:

Authors investigated only mRNA level of SCD and FADSs using qRT-PCR in three area samples of GBM. That means authors do not show the mechanism or the reason. Therefore, authors need to investigate in vitro experiment using GBM cell lines including U87 MG cell. For example, you may measure SCD and FADS mRNA level after necrotic agent treatment etc. You described in the conclusion section of abstract as ‘The expression of SCD and FADS2 was lower in the tumor than in the peritumoral area. This is just one of result, not conclusion. Therefore, you have to change the sentence. As you know, it is physiologically different between peritumaral area and normal area. If possible, you also have to compared with normal area. Authors described "It is likely that increased EPA production and further metabolism of this fatty acid are important for survival in GBM patients. For example, EPA may be metabolized to series 3 prostaglandins, e.g. PGE3, which is known for its anti-neoplastic properties as it competes with PGE2 for a receptor bond and thus mitigates the oncogenic action of PGE2 [71]. FADS1 and FADS2 are also involved in the production of EPA and 377 DHA, whose metabolites have anti-neoplastic properties [40,41,71]. Metabolism of these n-3 fatty acids in the brain is very intense [8,29,30] and therefore brain tumors may be expected to show a reduced expression of FADS1 and FADS2 compared to healthy brain tissue. Our results confirm this hypothesis, with their expression being lower in the GBM tumor than in the peritumoral area…….. in 4.3 section ( Expression of FADS1 and FADS2 in GBM tumors) of Discussion.Anti-cancer action of n-3 PUFA is important to n-6/n-3 PUFA ratio!!! As you know, FADSs produce Arachidonic acid as well as EPA and DHA. It means that the authors’ description in Discussion is very speculate. Therefore, you have to remove the section.

    5. If possible, you add GBM gross tissue photo with MRI scan photo for readers’ interpretation…..

Author Response

Rev.I.

In this review manuscript, authors described on ‘Expression of SCD and FADS2 is lower in the necrotic core and growing tumor area than in the peritumoral area of glioblastoma multiforme”. Although the results are partially original and interested in readers, there are many weakness and flaws that require authors’ further attention:

Major points:

Authors investigated only mRNA level of SCD and FADSs using qRT-PCR in three area samples of GBM. That means authors do not show the mechanism or the reason. Therefore, authors need to investigate in vitro experiment using GBM cell lines including U87 MG cell. For example, you may measure SCD and FADS mRNA level after necrotic agent treatment etc.

Our research shows that the expression of SCD and FADS2 is lower not only in the necrotic core but also in the growing tumor compared to the peritumoral area. Therefore, these differences are certainly not due to necrosis. Instead, they may be due to the fact that these areas differ in terms of quantitative and qualitative composition of different cells. In vitro tests are too simplified to explain some of the results observed in vivo.

You described in the conclusion section of abstract as ‘The expression of SCD and FADS2 was lower in the tumor than in the peritumoral area. This is just one of result, not conclusion. Therefore, you have to change the sentence.

The sentence was changed according to the reviewer's guidelines.

As you know, it is physiologically different between peritumaral area and normal area. If possible, you also have to compared with normal area.

Research on healthy human brain tissue is very controversial and therefore no research is conducted on such material. However, a study by Lemée et al. shows that the peritumoral area is the most suitable comparative control for brain tumour research.

Lemée, J.M.; Com, E.; Clavreul, A.; Avril, T.; Quillien, V.; de Tayrac, M.; Pineau, C.; Menei, P. Proteomic analysis of glioblastomas: what is the best brain control sample? J Proteomics 2013, 85, 165-173.

Authors described "It is likely that increased EPA production and further metabolism of this fatty acid are important for survival in GBM patients. For example, EPA may be metabolized to series 3 prostaglandins, e.g. PGE3, which is known for its anti-neoplastic properties as it competes with PGE2 for a receptor bond and thus mitigates the oncogenic action of PGE2 [71]. FADS1 and FADS2 are also involved in the production of EPA and 377 DHA, whose metabolites have anti-neoplastic properties [40,41,71]. Metabolism of these n-3 fatty acids in the brain is very intense [8,29,30] and therefore brain tumors may be expected to show a reduced expression of FADS1 and FADS2 compared to healthy brain tissue. Our results confirm this hypothesis, with their expression being lower in the GBM tumor than in the peritumoral area…….. in 4.3 section ( Expression of FADS1 and FADS2 in GBM tumors) of Discussion. Anti-cancer action of n-3 PUFA is important to n-6/n-3 PUFA ratio!!! As you know, FADSs produce Arachidonic acid as well as EPA and DHA. It means that the authors’ description in Discussion is very speculate. Therefore, you have to remove the section.

In accordance with the reviewer's recommendations, the indicated text has been deleted.

    5. If possible, you add GBM gross tissue photo with MRI scan photo for readers’ interpretation…..

According to the reviewer's comment, the photo has been enlarged.

Reviewer 2 Report

In this study the authors have determined the expression pattern of desaturase enzymes in different GBM tumor zones. Using mRNA analysis the distinct expression of enzymes is shown in distinguished tumor zones, however the study has some limitations that need to be addressed before considering for publication.

1) The authors are requested to provide the expression via immunohistochemistry of different desaturase enzymes in GBM tumor zones.

2) Is there any correlation with the enzyme expression on overall survival of GBM patients? The authors are requested to provide some evidence.

3) What is the expression in the normal brain tissue? It would be nice if authors can show the expression in normal epileptic brain control specimens

4) In fig 2. How many patients were analyzed to determine the expression?

5) Given the prevalence of IDH mutation in GBM, whether the mutational landscape of tumor affects the expression of these enzymes. The authors are requested to comment on this aspect in the discussion part.

6) The authors have provided contradictory evidence in fig 4. where the positive correlation of SCD, SCD5, FADS1, FADS2 was seen with one house keeping gene (B2M) but not with GAPDH. The authors should clearly explain this phenomenon by repeating this using third reference gene

7) The correlation of desaturase enzymes was examined using mRNA analysis in all three tumor zones. The authors are requested to provide some experimental evidence at protein level as well.

Author Response

Rev.II.

In this study the authors have determined the expression pattern of desaturase enzymes in different GBM tumor zones. Using mRNA analysis the distinct expression of enzymes is shown in distinguished tumor zones, however the study has some limitations that need to be addressed before considering for publication.

1) The authors are requested to provide the expression via immunohistochemistry of different desaturase enzymes in GBM tumor zones.

We are very grateful for the reviewer's comment, but unfortunately at the moment we do not have any material from patients that would allow us to perform immunohistochemical analysis. However, we plan to perform such analysis in the future.

2) Is there any correlation with the enzyme expression on overall survival of GBM patients? The authors are requested to provide some evidence.

Due to the small group of the examined patients, it is impossible to compare the published survival results with the level of expression of the examined desaturases. Therefore, we quoted survival results from the "human protein atlas" based on the cancer genome atlas (TCGA).

3) What is the expression in the normal brain tissue? It would be nice if authors can show the expression in normal epileptic brain control specimens

Research by Lemée et al. shows that the peritumoral area is the most suitable as a comparative control for brain tumour studies. That's why we used this control.

Lemée, J.M.; Com, E.; Clavreul, A.; Avril, T.; Quillien, V.; de Tayrac, M.; Pineau, C.; Menei, P. Proteomic analysis of glioblastomas: what is the best brain control sample? J Proteomics 2013, 85, 165-173.

4) In fig 2. How many patients were analyzed to determine the expression?

The number of patients is now given in the caption under the figure.

5) Given the prevalence of IDH mutation in GBM, whether the mutational landscape of tumor affects the expression of these enzymes. The authors are requested to comment on this aspect in the discussion part.

We are very grateful for this remark, but unfortunately the IDH analysis was not carried out during the hospital examinations of patients, and we do not have enough material to carry out such examinations.

6) The authors have provided contradictory evidence in fig 4. where the positive correlation of SCD, SCD5, FADS1, FADS2 was seen with one house keeping gene (B2M) but not with GAPDH. The authors should clearly explain this phenomenon by repeating this using third reference gene

In response to reviewer remark we have used third reference gene RPL13A. This gene is suggested as useful, stable internal control in glioblastoma. However, its expression is low and thus it is called “low abundance gene” [Madhuri et al “Validation of Housekeeping Genes for Gene Expression

Analysis in Glioblastoma Using Quantitative Real-Time Polymerase Chain Reaction”]. The authors show that expression level of RPL13A in glioblastoma samples averages 35 Ct and reject all the samples >40Ct.

We have used the same gene as reference and in our experiment the average was similarly high. However, the number of our samples was >40Ct, which makes them not eligible for further analysis (Fig.1). Therefore, despite our best efforts, we cannot use the RPL13A as internal control in our experiments.

Fig.1 Expression level of RPL13A in glioblastoma samples.

Nevertheless, literature data show that GAPDH is an appropriate reference gene for testing the expression of different genes in GBM tumors [Said et al. 2007; Kreth et al. 2010; Grube et al. 2015]. The expression of this enzyme is relatively the same in the brain and in GBM tumors. The level of B2M expression is different in brain and GBM tumor tissue [Kreth et al. 2010; Grube et al. 2015]. B2M expression in GBM tumors is also characterized by high heterogeneity. For this reason, we obtained different results for B2M and GAPDH.

The arguments cited above show that all the results obtained from desaturation expressions should be normalized only in relation to the GAPDH level. B2M should not be taken into account. Nevertheless, we decided to include data from B2M with a clear emphasis on the fact that it is a reference gene of low research value in research on GBM. We have done this in order to be able to compare it with other works in which someone else also used the B2M reference gene. Nevertheless, the aim of our work is not to test which reference gene is the best, but to analyze the expression of desaturase against a suitable reference gene.

Said HM, Hagemann C, Stojic J, Schoemig B, Vince GH, Flentje M, Roosen K, Vordermark D. GAPDH is not regulated in human glioblastoma under hypoxic conditions. BMC Mol Biol. 2007;8:55.

Kreth S, Heyn J, Grau S, Kretzschmar HA, Egensperger R, Kreth FW. Identification of valid endogenous control genes for determining gene expression in human glioma. Neuro Oncol. 2010;12(6):570-9. doi: 10.1093/neuonc/nop072.

Grube S, Göttig T, Freitag D, Ewald C, Kalff R, Walter J. Selection of suitable reference genes for expression analysis in human glioma using RT-qPCR. J Neurooncol. 2015;123(1):35-42. doi: 10.1007/s11060-015-1772-7.

7) The correlation of desaturase enzymes was examined using mRNA analysis in all three tumor zones. The authors are requested to provide some experimental evidence at protein level as well.

We would like to thank the Reviewer for this suggestion, but unfortunately the insufficient amount of material makes it impossible for us to conduct such research at the moment.

Reviewer 3 Report

Korbechi et al. analyze the mRNA expression of stearoyl-CoA desaturase and fatty acid desaturases in different areas of GBM. Although the paper is well written, the scientific novelty is questionable. In fact authors only analyze gene expression in these tumors. There are also some points that need to be improved. 

In the methods section, authors should better describe Table 1 and should include data on the genetic features of the GBM patients.  

In the results section, authors declare that their results change if they use different house-keepings. This result needs to be improved. Did the authors try another house-keeping? What happens if authors combine 2/3 house keeping genes? Furthermore, since these results are not clear, authors should protein expression of all these enzymes. 

Author Response

Rev.III.

Korbechi et al. analyze the mRNA expression of stearoyl-CoA desaturase and fatty acid desaturases in different areas of GBM. Although the paper is well written, the scientific novelty is questionable. In fact authors only analyze gene expression in these tumors. There are also some points that need to be improved.

In the methods section, authors should better describe Table 1 and should include data on the genetic features of the GBM patients.

We would like to thank the Reviewer for this recommendation, but unfortunately we do not have this data and so we can't present it.

In the results section, authors declare that their results change if they use different house-keepings. This result needs to be improved. Did the authors try another house-keeping? What happens if authors combine 2/3 house keeping genes?

In response to the Reviewer's remark we have used third reference gene RPL13A. This gene is suggested as useful, stable internal control in glioblastoma. However, its expression is low and thus it is called “low abundance gene” [Madhuri et al “Validation of Housekeeping Genes for Gene Expression

Analysis in Glioblastoma Using Quantitative Real-Time Polymerase Chain Reaction”]. The authors show that expression level of RPL13A in glioblastoma samples averages 35 Ct and reject all the samples >40Ct.

We have used the same gene as reference and in our experiment the average was similarly high. However, the number of our samples was >40Ct, which makes them not eligible for further analysis (Fig. 1). Therefore, despite our best efforts, we cannot use the RPL13A as internal control in our experiments.

Fig.1 Expression level of RPL13A in glioblastoma samples.

Nevertheless, literature data show that GAPDH is an appropriate reference gene for testing the expression of different genes in GBM tumors [Said et al. 2007; Kreth et al. 2010; Grube et al. 2015]. The expression of this enzyme is relatively the same in the brain and in GBM tumors. The level of B2M expression is different in brain and GBM tumor tissue [Kreth et al. 2010; Grube et al. 2015]. B2M expression in GBM tumors is also characterized by high heterogeneity. For this reason, we obtained different results for B2M and GAPDH.

The arguments cited above show that all the results obtained from desaturation expressions should be normalized only in relation to the GAPDH level. B2M should not be taken into account. Nevertheless, we decided to include data from B2M with a clear emphasis on the fact that it is a reference gene of low research value in research on GBM. We have done this in order to be able to compare it with other works in which someone else also used the B2M reference gene. Nevertheless, the aim of our work is not to test which reference gene is the best, but to analyze the expression of desaturase against a suitable reference gene.

Said HM, Hagemann C, Stojic J, Schoemig B, Vince GH, Flentje M, Roosen K, Vordermark D. GAPDH is not regulated in human glioblastoma under hypoxic conditions. BMC Mol Biol. 2007;8:55.

Kreth S, Heyn J, Grau S, Kretzschmar HA, Egensperger R, Kreth FW. Identification of valid endogenous control genes for determining gene expression in human glioma. Neuro Oncol. 2010;12(6):570-9. doi: 10.1093/neuonc/nop072.

Grube S, Göttig T, Freitag D, Ewald C, Kalff R, Walter J. Selection of suitable reference genes for expression analysis in human glioma using RT-qPCR. J Neurooncol. 2015;123(1):35-42. doi: 10.1007/s11060-015-1772-7.

Furthermore, since these results are not clear, authors should protein expression of all these enzymes.

We would like to thank the Reviewer for this recommendation, but due to the insufficient amount of material we cannot conduct such analysis.

Round 2

Reviewer 1 Report

Although the revised manuscript is improved compared to first manuscript, there are several weakness and flaws that require authors’ further attention: Minor points:

As I told you before, authors need to investigate in vitro experiment using GBM cell lines including U87 MG cell. For example, you may measure SCD and FADS mRNA level after necrotic agent treatment etc. If authors want to insist your conclusion, I think this data have to add in your results. I think that this is not difficult.... Although you changed the conclusion section of abstract, the sentence is not acceptable. For example: These results suggest that ----------------. A conclusion of Discussion section is also same thing.   As I told you before, you add GBM gross tissue photo with MRI scan photo for readers’ interpretation, not enlarge. Please add GBM gross tissue photo.

Author Response

Reviewer 1

Although the revised manuscript is improved compared to first manuscript, there are several weakness and flaws that require authors’ further attention: Minor points:

As I told you before, authors need to investigate in vitro experiment using GBM cell lines including U87 MG cell. For example, you may measure SCD and FADS mRNA level after necrotic agent treatment etc. If authors want to insist your conclusion, I think this data have to add in your results. I think that this is not difficult....

We are very grateful for this remark. In vitro tests have been added. We analyzed the effect of ROS (200 uM H2O2) on the expression of the tested desaturation in U87-MG cells. However, this model is not fully adequate for the condition inside the tumor. Because inside the GBM tumor there are small necrotic regions, which show poor vascularisation (a small number of blood vessels) [Gorin et al. 2004]. In these regions the neoplastic cells are in an environment with very small amount of nutrients. For this reason, to better reflect the conditions in the tumor, we have performed additional analyses. We investigated the effect of nutrient reduction on the expression of the tested desaturation on the U87-MG cell model. We reduced the concentration of sodium pyruvate and glutamine. However, the amount of glucose was unchanged in the medium. Glucose is converted into pyruvate and then into acetyl-CoA and further into fatty acids. Glutamine is also converted into acetyl-CoA and then into fatty acids [Ta et al. 2015]. In other words, we reduced the amount of substrate for the production of fatty acids and tested the expression of the studied desaturases. This model better represents the necrotic zones in the GBM tumor than the use of toxic substances that cause necrosis.

Ta NL, Seyfried TN. Influence of Serum and Hypoxia on Incorporation of [(14)C]-D-Glucose or [(14)C]-L-Glutamine into Lipids and Lactate in Murine Glioblastoma Cells. Lipids. 2015;50(12):1167-84. doi: 10.1007/s11745-015-4075-z.

Gorin F, Harley W, Schnier J, Lyeth B, Jue T. Perinecrotic glioma proliferation and metabolic profile within an intracerebral tumor xenograft. Acta Neuropathol. 2004 Mar;107(3):235-44.

Although you changed the conclusion section of abstract, the sentence is not acceptable. For example: These results suggest that ----------------. A conclusion of Discussion section is also same thing.  

Conclusions have been changed in line with the Reviewer's suggestions.

As I told you before, you add GBM gross tissue photo with MRI scan photo for readers’ interpretation, not enlarge. Please add GBM gross tissue photo.

We are very grateful for this remark, and we hope, that we have added an appropriate photo.

Reviewer 2 Report

Thanks for revised version of the manuscript and incorporating the comments, however, few minor concerns need to addressed. The authors did not comment on the IDH mutation in the discussion part. The authors are requested to add this in discussion section. Also, text editing and spelling errors need correction ( Page 6, line 200)

Author Response

Reviewer 2

Thanks for revised version of the manuscript and incorporating the comments, however, few minor concerns need to addressed. The authors did not comment on the IDH mutation in the discussion part. The authors are requested to add this in discussion section.

The appropriate fragment of discussion containing the effect of mutations in IDH1 / 2 genes on the expression of the desaturases tested has been added. However, we wanted to emphasize that mutations in IDH1 / 2 genes do not occur in primary GBM only in low grade glioma and in secondary GBM.

Also, text editing and spelling errors need correction (Page 6, line 200)

The manuscript has been checked again and the errors found have been corrected.

Reviewer 3 Report

Korbechi et al. improved their manuscript, however they do not answer to many of the questions raised by the reviewer. If authors are not able to perform protein analysis in western blotting of the tissue extracts, they should at least perform immunohistochemical analysis. In the present form the manuscript has not been improved enough.  

Author Response

Reviewer 3

Korbechi et al. improved their manuscript, however they do not answer to many of the questions raised by the reviewer. If authors are not able to perform protein analysis in western blotting of the tissue extracts, they should at least perform immunohistochemical analysis. In the present form the manuscript has not been improved enough.  

We perform protein analysis by immunohistochemistry method with SCD antibodies.